# Effects of Different Titanium Surfaces Created by 3D Printing Methods, Particle Sizes, and Acid Etching on Protein Adsorption and Cell Adhesion, Proliferation, and Differentiation

**DOI:** 10.3390/bioengineering9100514

**Published:** 2022-09-28

**Authors:** Max Jin, Haseung Chung, Patrick Kwon, Adil Akkouch

**Affiliations:** 1Western Michigan University Homer Stryker M.D. School of Medicine, Kalamazoo, MI 49008, USA; 2Department of Mechanical Engineering, College of Engineering, Michigan State University, East Lansing, MI 48824, USA; 3Medical Engineering Program, Department of Orthopedic Surgery, Western Michigan University Homer Stryker M.D. School of Medicine, Kalamazoo, MI 49008, USA

**Keywords:** 3D printing, surface topography, titanium disc, particle size, protein adsorption, cell adhesion, proliferation, differentiation, osseointegration

## Abstract

The surfaces of 3D printed titanium prostheses have major impacts on the clinical performance of the prostheses. To investigate the surface effects of the products generated by 3D printed titanium on osseointegration, six surface types of titanium discs produced by the direct metal laser sintering (DMLS) and electron beam melting (EBM) methods, with two sizes of titanium particles and post-printing acid etching, were used to examine the surface topography and to explore the protein adsorption, pro-inflammatory cytokine gene expressions, and MC3T3-E1 cell adhesion, proliferation, and differentiation. The EBM-printed disc showed a stripy and smooth surface without evidence of the particles used, while the DMLS surface contained many particles. After acid etching, small particles on the DMLS surface were removed, whereas the large particles were left. Moreover, distinct proteins with low molecular weights were attached to the 3D printed titanium discs but not to the pre-printing titanium particles. The small titanium particles stimulated the highest TNF-α and IL-6 gene expressions at 24 h. The alizarin red content and osteocalcin gene expression at day 21 were the highest in the groups of acid-etched discs printed by DMLS with the small particles and by EBM. Therefore, the acid-treated surfaces without particles favor osteogenic differentiation. The surface design of 3D printed titanium prostheses should be based on their clinical applications.

## 1. Introduction

By 2030, around 572,000 whole hips and 3.48 million whole knees are expected to be replaced each year in the US, and it is estimated that about 96,700 hip and 268,200 knee revision surgeries will be conducted annually [1]. The revision surgeries put a heavy burden on patients and society [2]. The most common cause for revision surgery is aseptic loosening [3,4], which is considered to result from wear debris, stress shielding, and micromotion at the bone-implant interface [5].

One of the key challenges in the artificial joint field is how to manufacture the customized artificial joint to mimic the biological counterpart in its morphology, structures, mechanics, and biology. Such a customized artificial joint must be capable of transmitting loading to the bone uniformly and within the physiological threshold [6,7]. During the past few decades, the additive manufacturing (AM or 3D printing) has been applied in the medical field. Compared to conventional procedures such as casting or machining, AM offers a greater design flexibility in the fabrication of complex prostheses without tooling and molds [8,9]. As multiple different prostheses can be printed in one batch, AM may reduce waste, processing cost, and time [8]. Therefore, AM is more suitable for producing customized artificial prostheses to deal with various clinical situations [10].

The two most common powder bed fusion (PBF) AM technologies are direct metal laser sintering (DMLS) and electron beam melting (EBM). Titanium (Ti) alloy prostheses fabricated by the conventional methods such as casting and machining have been shown to have good biocompatibility and corrosion resistance [11,12,13,14]. To summarize, Ti products made by AM have also demonstrated good biocompatibility and corrosion resistance [15,16,17,18,19]. A multicenter clinical trial for 3D printed single-Ti dental implants revealed 94.5% of a 3-year survival rate [20]. The Ti-6Al-4V dental implant made by DMLS with the powders of the diameter 25–45 µm had 66.1 ± 4.5% bone-to-implant contact in five years [21]. In addition, 3D printed Ti scaffolds have been used to reconstruct the large osseous defects in the ankle and foot [22], as well as to replace femoral heads with osteonecrosis [23]. More encouragingly, the FDA has approved a 3D printed Ti iFuse-3D implant to be used clinically for the replacement of the human sacroiliac joint. The current clinical performance of 3D printed Ti prostheses presents a huge potential due to their wide applications in solving complicated clinical issues and in improving clinical outcomes.

Osseointegration is the key factor in determining bony prosthesis clinical performance [24,25]. Prosthesis surface properties such as topography have huge impacts on osseointegration [24,26,27]. The surface created by both PBF AM technologies contains many partially sintered powder particles. The amount of the particles on the surface depends on the 3D printing parameter setting and the energy sources [28]. However, there are controversial opinions on whether the particles on the surface should be removed. One concern is that the particles falling off from the surface as wear debris may cause the aseptic loosening of the bony prosthesis [29,30,31]. On the other hand, some researchers vote for keeping the particles on the 3D printed bony prosthesis surfaces because the partially sintered particles form a rough surface, which is favorable to bone ingrowth and thus enhances osseointegration [32]. Therefore, the effects of the surface topography of 3D printed Ti implants on bone formation should be clarified before their application.

The purpose of this study is to investigate the surface effects of the 3D printed Ti discs on osseointegration. Altogether, six surface types of Ti discs were created with DMLS and EBM, two sizes of the Ti6Al4V powders (average 22.5 and 100 µm in diameter), and post-printing acid etching. They were examined for protein adsorption, MC3T3-E1 cell adhesion, proliferation, and osteogenic differentiation.

## 2. Materials and Methods

### 2.1. Fabrication of 3D Printed Ti Discs

In this study, two AM methods, DMLS and EBM, were used to create Ti discs. The main difference between DMLS and EBM lies in the energy source. As an energy source to melt metal powders, a laser is used by DMLS while an electron beam is used by EBM. Moreover, DMLS works in the air, whereas EBM is performed in a vacuum chamber. Furthermore, the metal powder for EBM needs preheating, while the metal powder for DMLS does not. The small particle powder Ti6Al4V (3D Systems Corp., Rock Hill, SC, USA) with an average size of 22.5 µm (5–40 µm) and the large particle powder Ti6Al4V (Oerlikon Metco Inc. Westbury, NY, USA) with an average size of 100 µm (45–150 µm) were used by DMLS, while only the large particle powder Ti6Al4V (average size of 100 µm) was used by EBM due to the EBM requirement. The detailed 3D printing procedures and conditions were reported by our group before [33]. The specific processing parameters used while fabricating parts with DMLS technology were 190 W laser power, 1600 mm/s scan speed with 70 μm hatch spacing, and 30 μm layer thickness. Three types of Ti discs with different surface topographies were produced:DMLS/SP: the disc printed by DMLS with small particle powders Ti6Al4V (average size of 22.5 µm);DMLS/LP: the disc printed by DMLS with large particle powders Ti6Al4V (average size of 100 µm);EBM/LP: the disc printed by EBM with large particle powders Ti6Al4V (average size of 100 µm).

Here, SP stands for small particle, whereas LP stands for large particle.

To explore the effects of etching in acid on the surface properties of 3D printed Ti discs, some discs from each group were treated with acid etching. The printed discs were etched using 37% hydrochloric acid (HCL) and 97% sulfuric acid (H_2_SO_4_) at room temperature for 3 h. They were then placed under 97% H_2_SO_4_ for another 2 h, rinsed thoroughly in deionized water afterwards, and dried in air. Accordingly, 3 acid etching groups were produced: Acid_DMLS/SP, Acid_DMLS/LP, and Acid_EBM/LP. The 3D printed Ti discs were about 4.2 cm in diameter and 2 mm in thickness.

### 2.2. Scanning Electron Microscopy (SEM), Energy Dispersive X-ray Spectroscopy (EDX), and Surface Roughness

#### 2.2.1. SEM and EDX Analysis

The scanning electron microscope (PROX DMP 200, 3D Systems Corp., Rock Hill, SC, USA) was utilized to visualize the surface morphologies of the specimens. In this study, 6 types of 3D printed Ti discs were analyzed without prior gold sputter coating and visualized using 20 kV accelerating voltage so that the electron detector could receive stronger and clearer signals. In the meantime, the chemical element composition of the disc from each group and pre-printing of the small and the large particles was measured by EDX.

#### 2.2.2. Surface Roughness

The surface roughness of the 3D printed Ti discs using DMLS and EBM, before and after acid etching, was measured using a 3D laser scanning microscope LEXT OLS4000 (Olympus Corporation of America. Breinigsville, PA, USA).

### 2.3. Protein Adsorption

Three discs from each group were cleaned by the ultrasonic cleaner Branson 1800 (Branson Ultrasonics Corporation, Brookfield, CT, USA) for 15 s to remove the Ti particles from the disc surfaces. Then, the discs were placed in 70% ethanol for 1 h at room temperature and rinsed with aseptic phosphate buffered saline (PBS) (Sigma-Aldrich, St. Louis, MO, USA) three times. Subsequently, each of the cleaned and disinfected discs and 10 mg of each size of the particles were placed into 6-well plates and incubated in α-MEM medium (HyClone Laboratories, Logan, UT, USA) containing 10% fetal bovine serum (FBS) (HyClone Laboratories, Logan, UT, USA) and antibiotics (100 U/mL penicillin and 100 U/mL Streptomycin) (ThermoFisher Scientific, Waltham, MA, USA) at 37 °C overnight.

After overnight incubation, the discs and the powder particles were taken out and rinsed with PBS 3 times. Then, the proteins adsorbed on the surface of each disc were washed off using 200 µL 3M urea repeatedly. The solutions of the 3 discs in each group were pooled together, lyophilized, and re-dissolved using 100 µL MilliQ water. The proteins adsorbed on the surfaces of the small and large powder particles were washed off using 600 µL 3M urea, lyophilized, and re-dissolved in 100 µL MilliQ water.

The protein contents in each group were quantified using Pierce™ BCA Protein assay (ThermoFisher Scientific, Waltham, MA, USA) according to the manufacturer’s instructions. Then, 2 µg protein of each sample in the loading buffer was added into 4–20% Precise™ Protein Gels (ThermoFisher Scientific, Waltham, MA, USA) to run SDS-PAGE electrophoresis. After electrophoresis, the gel was stained using Precise™ Silver Stain (ThermoFisher Scientific, Waltham, MA, USA) according to the manufacturer’s instructions.

### 2.4. Cell Adhesion

Three discs from each group were cleaned, disinfected, and incubated overnight in 6-well plates with α-MEM medium containing 10% FBS and antibiotics, as detailed above. Then, 2 × 10^5^ MC3T3-E1 cells were added into each well with the disc. According to a previous report [34], we adopted a 24-h time point. After 24 h, each disc was rinsed by PBS 3 times to remove the unattached cells and then incubated in 0.25 mg/mL (3-(4, 5-dimethyl-2-thiazolyl)-2, 5-diphenyl-2H tetrazolium bromide) (MTT) (Sigma-Aldrich, St. Louis, MO, USA) in PBS solution at 37 °C and 5% CO_2_ [35]. After 3 h, each disc was rinsed twice with PBS and incubated at room temperature in 2 mL dimethyl sulfoxide (DMSO) (Sigma-Aldrich, St. Louis, MO, USA) for 1 h. Then, the MTT concentration was measured at 570 nm, which is directly proportional to the number of viable cells on the discs.

### 2.5. Cell Proliferation

Three discs from each group were cleaned, disinfected, and incubated overnight in 6-well plates with α-MEM medium containing 10% FBS and antibiotics, as detailed above. Then, 1 × 10^5^ MC3T3-E1 cells were added into each well with the disc. The α-MEM medium containing 10% FBS and the antibiotics in 6-well plates were changed every two days. At day 14, each disc was rinsed by PBS twice and incubated in 0.25 mg/mL MTT in PBS solution at 37 °C and 5% CO_2_ for 3 h. Then, each disc was rinsed twice with PBS and incubated at room temperature in 2 mL DMSO for 1 h. The MTT concentration in the DMSO was measured at 570 nm.

### 2.6. Osteocalcin Gene Expression and Alizarin Red Measurement

#### 2.6.1. Osteocalcin Gene Expression

Three discs from each group were cleaned, disinfected, and incubated overnight in 6-well plates with α-MEM medium containing 10% FBS and antibiotics, as detailed above. Then, 2 × 10^5^ MC3T3-E1 cells were added into each well with the disc and cultured in the mineralization medium (α-MEM medium containing 10% FBS, 100 U/mL penicillin and 100 U/mL Streptomycin, 50 µg/mL ascorbic acid, and 10 mM β-glycerophosphate) (Sigma-Aldrich, St. Louis, MO, USA). The mineralization medium was changed every two days.

At day 21, the discs were rinsed with cold PBS twice, and the total RNA was extracted using the RNeasy Mini Kit (Qiagen, Germantown, MD, USA) according to the manufacturer’s instructions. The concentration and purity of the total RNA were quantified using the NanoDrop One Microvolume UV-vis Spectrophotometers (ThermoFisher Scientific, Waltham, MA, USA). One microgram of the extracted RNA from each sample with random hexamers was used to perform the reverse transcription reaction (RT) with a TaqMan Reverse Transcription reagent kit (ThermoFisher Scientific, Waltham, MA, USA) to synthesize cDNA according to the manufacturer’s instructions. The osteocalcin (OCN) gene expression of the MC3T3-E1 cells cultured on each type of the 3D printed Ti discs was measured by real-time quantitative PCR (qRT-PCR).

qRT-PCR was performed using TaqMan^®^ Universal PCR Master Mix (Applied Biosystems, Waltham, MA, USA) in an ABI-7500 Real-Time PCR System (ThermoFisher Scientific, Waltham, MA, USA). Briefly, 30 µL PCR reaction was prepared with 2 µL cDNA and 1.5 µL gene specific probe and 15 µL master mix from TaqMan^®^ Gene Expression Assays (Applied Biosystems, Woburn, MA, USA). The assay IDs of the probes used are listed in Table 1. The relative quantification of the target genes was calculated by the 2^−∆∆Ct^ method [36], and GAPDH was used as a housekeeping gene for the normalization of the data.

#### 2.6.2. Alizarin Red Staining

To evaluate the mineralization by MC3T3-E1 cells seeded on 3D printed Ti discs, the calcium deposition was evaluated using alizarin red staining (ARS). The discs seeded with cells were cultured in mineralization medium for 21 days. For calcium staining, the discs were washed with MilliQ water 3 times and stained with 2% alizarin red (pH 4.1) solution overnight (Sigma-Aldrich, St. Louis, MO, USA). Then, the discs were rinsed with MilliQ water until the MilliQ water became colorless and dried in air. Each of the dried discs was immersed in 2 mL of 1N HCl solution overnight with shaking. The concentration of the extracted ARS from each disc was measured at 405 nm.

### 2.7. TNF-α and IL-6 Genes Expression

Three discs from each group were cleaned, disinfected, and incubated overnight in a well of the 6-well plates with α-MEM medium containing 10% FBS and antibiotics, as detailed above. Then, 2 × 10^5^ MC3T3-E1 cells were added into each well with the disc. On the other hand, both the small and the large Ti particles were added to wells containing confluent MC3T3-E1 cells in 6-well plates. The final concentration of the particles at each well was 1 mg/mL. The cells were cultured at 37 °C and 5% CO_2_. After 1 and 21 days, the cultures were rinsed 3 times with PBS (1×). RNA extraction from each sample was conducted using the RNeasy Mini Kit (Qiagen, Germantown, MD, USA) according to the manufacturer’s manual. The TNF-α and IL-6 gene expressions of each sample were measured similarly by qRT-PCR, as described above. QuantStudio3 (ThermoFisher Scientific, Waltham, MA, USA) was used in place of the ABI-7500. The assay IDs of the probes used are listed in Table 1.

### 2.8. Statistics Analysis

The data in this study were analyzed by ANOVA and compared by Tukey’s method using GraphPad Prism 8 (GraphPad Software, San Diego, CA, USA).

## 3. Results

### 3.1. Ultramorphology, Chemical Composition, and Surface Roughness

#### 3.1.1. SEM and EDX Analysis

Figure 1 showed the SEM images of six types of the 3D printed Ti discs. The EBM/LP group had a porous and woven stripy surface. The stripy surface was smooth, and the particles were hardly seen, which indicates that the EBM printing resulted in the near complete melting and fusion of the Ti particles. However, DMLS/LP and DMLS/SP exhibited a rough surface containing a lot of particles. Large particles were found in the DMLS/LP group while small particles were found in the DMLS/SP group. These particles matched the original two Ti particle sizes of 22.5 and 100 µm in diameter, respectively. After acid etching, the surface looked smoother in the Acid_EBM/LP group. In the Acid_DMLS/LP group, the small particles were removed from the surface while the large ones were not. Many tiny dents were left on the surface, even on the surfaces of the remaining large particles. In the Acid_DMLS/SP group, the particles were completely washed off from the surface. Only the dents left on the surface could be seen. The dents were the areas where the small particles were fused into the other particles when they were partially melted by laser.

Figure 2 presented the EDX spectrums of all eight groups. The elements of Ti, aluminum (Al), vanadium (V), oxygen (O), and carbon (C) were detected in all groups. The element of iron (Fe) was seen in the SP and DMLS/SP groups. Table 2 gives the atomic percentage of each element detected in the EDX spectrum. The Ti element accounted for about 80%, and the other elements made up from 4 to 10%. The trace amount of iron in the SP and DMLS/SP groups was around 0.5 to 0.8%. There was no significant difference in the element composition among all the groups, nor did the acid etching change the element composition.

#### 3.1.2. Surface Roughness

Figure 3 demonstrates the surface roughness measurement results of all 3D printed Ti discs. Before acid etching, the DMLS/SP disc had the smoothest surface compared to the two EBM/LP and DMLS/LP ones (*p* < 0.01). There was no significant difference between the EBM/LP and the DMLS/LP groups. After acid etching, the EBM/LP group had the roughest surface among the three acid-etched groups (*p* < 0.01). When comparing the surface roughness before and after the acid etching, no surface roughness changes were found in the DMLS/SP and DMLS/LP discs. However, the Acid_EBM/LP disc surface became rougher than the EBM/LP one (*p* < 0.01). This roughness measurement should be interpreted carefully because the roughness Ra came from both the surface roughness and the fused particle height.

### 3.2. Protein Adsorption

To investigate the effects of the 3D printing methods, powder sizes, and acid etching on the protein adsorption, we extracted the FBS proteins adsorbed on the surfaces of the small particle powder, large particle powder, DMLS/SP, DMLS/LP, EBM/LP, Acid_DMLS/SP, Acid_DMLS/LP, and Acid_EBM/LP discs and ran the SDS-PAGE gel.

As shown in Figure 4, the small and large Ti particles had the same stained bands before the DMLS and EBM processing. However, the 3D printed discs by DMLS and EBM with the particles of varying sizes had different adsorbed proteins. The SDS-PAGE gel showed that the proteins between 15 KDa and 60 KDa had obviously different band sizes and staining intensities, which means the proportion of each corresponding protein in the total protein was different. The small as well as the large particles showed larger bands corresponding to larger molecular weight (above 50 KDa). In addition, the proteins located between 15 and 40 KDa were seen only on the discs, but not on the particles. Furthermore, the proteins located between 25 and 30 KDa were found only on the discs fabricated by DMLS, but not by EBM. On the other hand, no significant difference in the stained bands was seen before and after acid etching on the 3D printed discs by EBM. The bands at 25 KDa were seen on the discs printed by DMLS. After acid etching, these bands disappeared, and a new protein band at around 15 KDa appeared with a strong stained intensity. These changes were seen on the 3D printed discs by DMLS with both the small and the large particles. In addition, more intense bands were seen for the acid-treated discs made by the small particles than those made by the large particles.

### 3.3. Cell Adhesion and Proliferation

Figure 5A presents the adhesion of the MC3T3-E1 cells at 24 h to the six types of the Ti discs measured by the MTT assay. There were no statistically significant differences in the cell adhesion among all the groups except the two groups of DMLS/SP and DMLS/LP. The surface created by the DMLS with large particles had more attached cells than the surface created by the DMLS with small particles (*p* < 0.01). In particular, there was no difference before and after the acid etching.

At day 14, the MTT measurement showed that there was no difference in the proliferation among all the groups (Figure 5B).

### 3.4. TNF-α and IL-6 Gene Expressions

The gene expressions of the TNF-α and IL-6 of the MC3T3-E1 cells cultured on six types of 3D printed Ti discs with small and large particles for 24 h were quantified by real-time qPCR using the 2^−ΔΔCt^ method and normalized against DMLS/SP (Figure 6). The TNF-α gene expression was the highest in the non-printed small particle group (Figure 6A). It was approximately more than 7-fold higher than the other groups. The DMLS 3D printing resulted in the decrease in the TNF-α expression when the small particle powder was used as the starting material. When the large particle powder was used for the printing, the DMLS technology had no effect on the expression of TNF-α. On the other hand, the EBM process resulted in a decrease in the TNF-α expression. There was no statistically significant difference among the other printed discs. Similarly to the TNF-α expression, the non-printed small particles showed the highest IL-6 expression when compared to the non-printed large particles and to the 3D printed discs (Figure 6B). The DMLS printing resulted in the lowest IL-6 expression when the small particle powder was used. Before acid etching, the DMLS printing with small particles showed less IL-6 expression than those of the DMLS and EBM printing with large particles. After acid etching, the EBM printing resulted in greater IL-6 expression than the DMLS printing. Interestingly, acid etching caused an increase in IL-6 gene expression in the DMLS/SP discs, while it caused a decrease in the DMLS/LP discs when compared to the non-treated DMLS discs. IL-6 gene expression level had no change in the EBM printing groups before and after acid etching.

As expected, at the 21-day cell culture there were no statistically significant differences in the TNF-α and IL-6 gene expressions among all the 3D printed Ti discs (Figure 7).

### 3.5. Mineralization and Nodule Formation

Osteocalcin (OCN) is a marker to indicate functional osteoblast formation [37]. The OCN gene expression in each group is shown in Figure 8A. The post-treatment using acid etching resulted in increased OCN gene expressions in the Acid_DMLS/SP and Acid_EBM/LP discs compared to the other printed discs. Paralleled with the OCN gene expression, they also had more calcium deposition (alizarin red contents) when compared to all untreated discs as well as the Acid_DMLS/SP discs (Figure 8B), signaling that these two groups had more calcium in the mineralization than the other groups.

## 4. Discussion

In this study, we discovered that the 3D printing methods and powder particle sizes created distinct surface properties. Other researchers demonstrated similar results [38,39,40,41]. Interestingly, this study further revealed that the effects of acid etching on the surface of 3D printed Ti discs have correlations with the remaining particle sizes. The acid etching procedure used in this study was able to only remove the small particles (average 22.5 µm), not the large particles (average 100 µm) (Figure 1), which may be determined by the sintered areas between the particles. If the particles were completely fused (melting), acid etching would hardly change the surface topography on the EBM group. If the particles were partially sintered, only the sintered areas formed by the small particles were affected by acid etching. Certainly, the sintered areas are controlled by the 3D printer settings. In other words, the sintered areas are decided by the energy applied onto the Ti particles [28]. Furthermore, the acid etching effects are also related to the concentration and composition of the acid used. Different concentrations and compositions of the acid may have their distinct effects on the sintered areas between the particles [42]. Thus, it is necessary to adopt the appropriate acid etching system based on different needs.

It is reported that the rough surfaces created by 3D printing had greater osseointegration because the fixation strength of the rough-textured implants provided superior interlocking relative to the smooth-textured ones [32]. The surface roughness measurement (Figure 3) showed that the Ti discs printed by EBM and DMLS with a large particle powder had rougher surfaces compared to those printed by DMLS with a small particle powder. Therefore, the surfaces with large particles (average 100 µm) or no particles, such as those of the EBM discs, may be better for osseointegration than the surfaces with small particles.

Protein adsorption to an implant surface is considered as the first event that happens when the implant is placed into the bone osteotomy. The amount, composition, and stereo configurations of the adsorbed proteins affect cell attachment, proliferation, differentiation, maturation, and osteointegration [43,44,45]. Figure 4 shows that there was no difference in the protein adsorption to the surface of the pre-print Ti powder particles, which indicates that the powder particle size itself does not influence the composition of the adsorbed proteins. After the 3D printing process, small molecular weight proteins were adsorbed on the surfaces of the Ti discs. This result demonstrated that both DMLS and EBM may have changed the Ti surface properties, and consequently, the 3D printed Ti implants could have different proteins attached. After the acid etching treatment, no obvious changes in the protein bands were seen in the Ti discs printed with EBM before and after acid treatment. However, there were differences in the protein adsorption in the Ti discs printed with DMLS, which is parallel with the observations in the SEM images (Figure 1). Acid etching removed many small particles in the DMLS discs and left many tiny dents, but such changes were not found in the acid-treated EBM discs. This surface topography may have changed the surface microtexture, energy, and charges, thus affecting different protein adsorptions. The EDX analysis revealed that the element composition of Ti, Al, V, O, and C in the surface of the disc was not changed with the 3D printing and acid etching. The identification of the proteins and their concrete effects on the cell biological activities also need to be further explored.

The implant surface plays very important roles in cell attachment, migration, proliferation, and differentiation. There are three major considerations regarding its mechanisms. First, we found that different printing processes, as well as different Ti particle sizes, resulted in different protein adsorptions on the surface of the fabricated implants. It is well known that the adsorbed proteins have different effects on cell activities, such as adhesion, proliferation, and differentiation [46], as well as implant osseointegration. Secondly, the adsorbed proteins on the implant surface recruit inflammatory cells in situ, especially macrophages, to release cytokines. Some of them are pro-inflammatory cytokines or chemokines, which aggravate inflammation and damage tissues [47] and which may lead to implant failure. Alternatively, some of them play critical roles in stem cell recruitment, proliferation, and differentiation, which enhances the repair process and osseointegration of the implant with the native tissue [48]. Thirdly, mechanical properties such as stiffness affect cell migration, proliferation, and differentiation [49]. This study demonstrated that even though different surfaces did not directly bring about the significant changes in cell attachment and proliferation, they had different protein adsorptions and induced pro-inflammatory TNF-α and IL-6 gene expressions at different levels. The acid etching and the 3D printing processes have introduced distinct proteins with low molecular weight attached to the different surfaces. Further studies on the protein identification, the exact binding sites on the different surfaces, and their effects on cell differentiation are required. More research should also be carried out to detect how the small particles triggered the highest TNF-α and IL-6 gene expressions of the MC3T3-E1 cells presented in our study, enabling us to better understand the mechanism by which the fall-off particles may cause osteolysis around implanted 3D artificial joints, resulting in implant failure.

More interestingly, the Acid_DMLS/SP and Acid_EBM/LP groups favored MC3T3-E1 cell mineralization more than the other groups. These two groups shared one common characteristic, which was that they had no particles on their surface. This result should be interpreted carefully. Osseointegration not only means bone formation on the surface of the implant, it also needs bone ingrowth into the rough and porous surface to form the interlock with the implant. As a result, the implant can be anchored firmly into the native bone, which limits micromovements and increases the lifespan of the joint replacements. From this point, the acid-etched EBM-printed surface may be more favorable to osseointegration.

IL-6 and TNF-α are important cytokines that are considered as predictors to indicate implant loosing or failure [50,51]. Here, our study demonstrated that the pre-print small particles stimulated the MC3T3-E1 cells to increase their TNF-α gene expression about 7-fold and their IL-6 gene expression about 1.3 to 3.3 times, which indicated that free small particles might cause more inflammatory reactions. In addition, higher IL-6 gene expression was found in the DMLS/LP discs than in the DMLS/SP discs, and there was no difference between the EBM/LP and the DMLS/LP discs. This result revealed that the 3D printing methods yielded little impact on the pro-inflammatory cytokine release and that particle sizes indeed affected the pro-inflammatory cytokine production. A non-significant change in IL-6 gene expression was found in the EBM-printed discs before and after acid etching. This is because of the EBM-printed discs’ lack of particles on their surface due to the melting process during 3D printing. As shown in Figure 1, the Ti particles were completely fused. Indeed, they were hardly affected by the acid etching treatment. In contrast, the acid etching removed small particles from the surfaces of DMLS/SP- and DMLS/LP-printed discs, as shown in Figure 1; so, the released pro-inflammatory cytokines could have been altered.

However, significant differences in the IL-6 and TNF-α gene expressions were not seen in the MC3T3-E1 cells cultured for 21 days on the different Ti discs. It may be related to the extracellular matrix formation. The MC3T3-E1 cells might not produce enough extracellular matrix at 24 h to cover the 3D printed Ti discs. Obviously, surface topography plays a key role in cell responses. In a 21-day culture, the MC3T3-E1 cells formed a thick layer of extracellular matrix, which covered the whole surface of the Ti disc. At this moment, the MC3T3-E1 responded mainly to the extracellular matrix rather than to Ti disc surfaces.

One of the mechanisms considered to cause the aseptic loosing and periprosthetic osteolysis of hip or knee artificial joints was that the pro-inflammatory cytokines secreted by macrophages phagocytizing the wear debris triggered bone loss [4]. The effective wear debris size for phagocytosis is between 0.1 and 10 µm [31,52]. Kaluderovic et al. showed that human osteoblasts also have autophagy functions, and autophagy affects osteoblast functions [53]. However, the exact mechanisms of the pre-osteoblastic MC3T3-E1 reactions to the different surfaces and the Ti particles of both sizes in this study are not fully clear.

## 5. Conclusions

The surface properties of 3D printed prosthesis can be manipulated by the 3D printing procedure, the particle size, and the post-printing acid treatment. EBM created a porous and woven stripy surface without particles, while DMLS produced a surface with many particles. The size of those particles corresponded to the size of the used Ti particles. The post-printing acid etching treatment, used in this study, removed the small particles from the surfaces of the DMLS-printed Ti discs but not the large particles. Surface roughness is decided by the particle size but not by the 3D printing methods. Large particles created a rougher surface. The acid etching treatment used in this study only made the EBM disc surface rougher, but not the DMLS ones. In addition, the 3D printing method and acid etching did not change the surface element composition of Ti, Al, V, O, and C.

Such modifications to the surface properties of 3D printed Ti prosthesis affected the protein adsorption and the cell biological activities. The EBM- and DMLS-printed Ti surfaces attracted low molecular weight proteins, which was not seen in the pre-printing particles. The post-printing acid treatment also changed the composition and the amount of the proteins adsorbed on the surfaces. The surface properties of the Ti discs also played a role in the pro-inflammatory gene expressions by the MC3T3-E1 cells. At day 1, the pre-printing small particles caused the highest IL-6 and TNF-α gene expressions. The surfaces of the 3D printed Ti discs with large particles increased the IL-6 expression compared to the ones with small particles. At day 21, there were no differences in the IL-6 and TNF-α gene expressions in all the groups. Interestingly, the TNF-α gene expressions increased, while the IL-6 gene expressions decreased at 21 days compared to those at day 1. Furthermore, the surface properties of the Ti discs had significant impacts on the MC3T3-E1 cell differentiation but not on the adhesion and proliferation. At 21 days, the surfaces of the acid-treated Ti discs printed by EBM with large particles and by DMLS with small particles had more calcium deposition and OCN gene expression, which indicated that those surfaces favor MC3T3-E1 cell mineralization in vitro.

In summary, different 3D printed implant surfaces should be selected based on their clinical applications. The dental implant body excluding the neck area may require certain particles to increase surface roughness and obtain more bony interlock to improve osseointegration. Wear debris is not a major issue here because no heavy force is put on a dental implant. On the other hand, worn-off particles, which bring about inflammation and bone adsorption, may be fatal to the survival rate of artificial joints such as those of the hip or knee. Therefore, prostheses with smooth surfaces instead of the ones with various particles should be chosen to avoid the peri-prosthesis osteolysis induced by the fall-off particles and, accordingly, to improve their clinical endurance and performance.

## Figures and Tables

**Figure 1 bioengineering-09-00514-f001:**
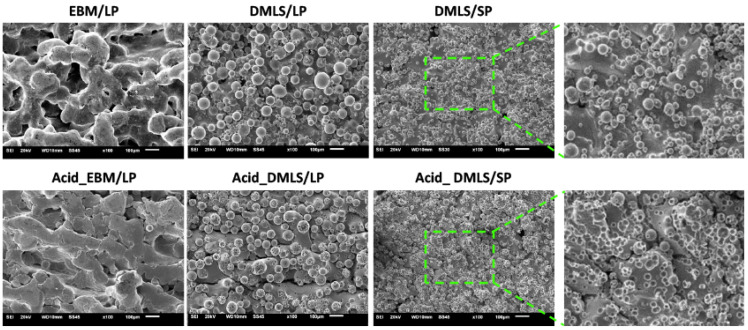
SEM images of 6 types of 3D printed Ti discs. The EBM/LP group showed porous, woven stripy surface. The stripes were smooth. DMLS-printed discs displayed particle surfaces. After acid etching, no obvious changes were found in the Acid_EBM/LP group. The small particles were removed from the Acid_DMLS/LP group. The large particles still existed. Many dents were left on the surfaces of the large particles after the removal of the small particles by acid etching. In the Acid_DMLS/SP group, all the particles disappeared. Only the small dents were left. The photos in right panel are the enlarged ones of the green squares. DMLS: direct metal laser sintering. EBM: electron beam melting. LP: large particle, SP: small particle. Scale bars: 100 µm.

**Figure 2 bioengineering-09-00514-f002:**
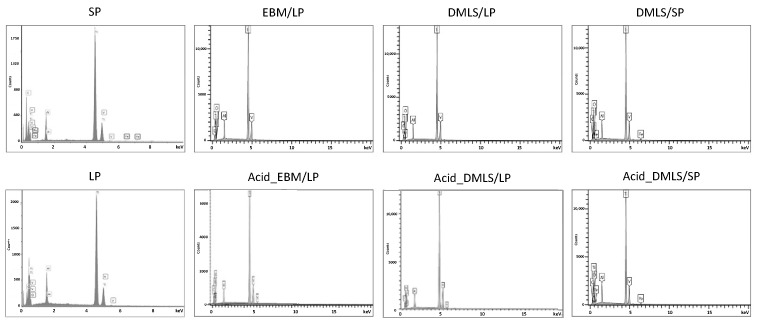
EDX spectrum of each group. The elements of Ti, aluminum, vanadium, oxygen, and carbon were found in all the groups. The iron element was detected only in the SP and DMLS/SP groups.

**Figure 3 bioengineering-09-00514-f003:**
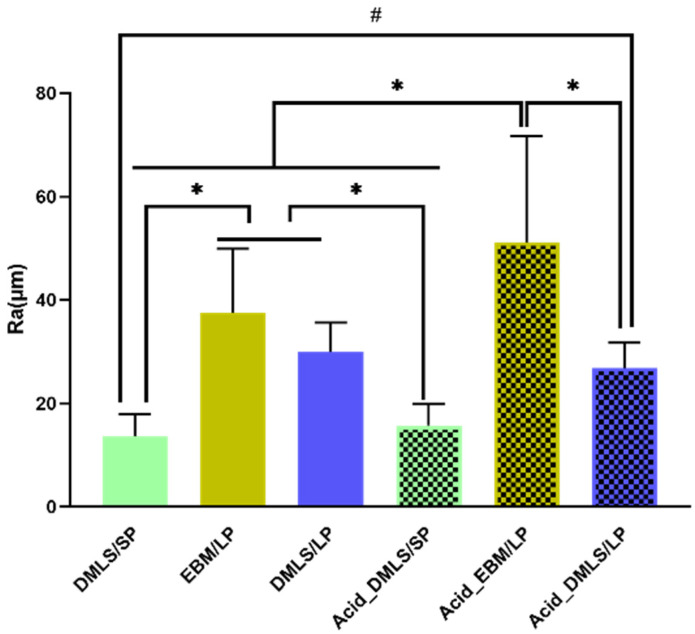
Surface roughness measurement of Ti discs of each group. Before acid etching, the DMLS/SP disc had the smoothest surface compared to the two EBM/LP and DMLS/LP ones. After acid etching, the EBM/LP group had the roughest surface among the three acid-etched groups. When comparing surface roughness before and after acid etching, no surface roughness changes were found in the DMLS/SP and DMLS/LP discs. However, the Acid_EBM/LP disc surface became rougher than the EBM/LP one (*p* < 0.01). DMLS: direct metal laser sintering. EBM: electron beam melting. LP: large particle, SP: small particle. * indicates *p* < 0.01; # indicates *p* < 0.05.

**Figure 4 bioengineering-09-00514-f004:**
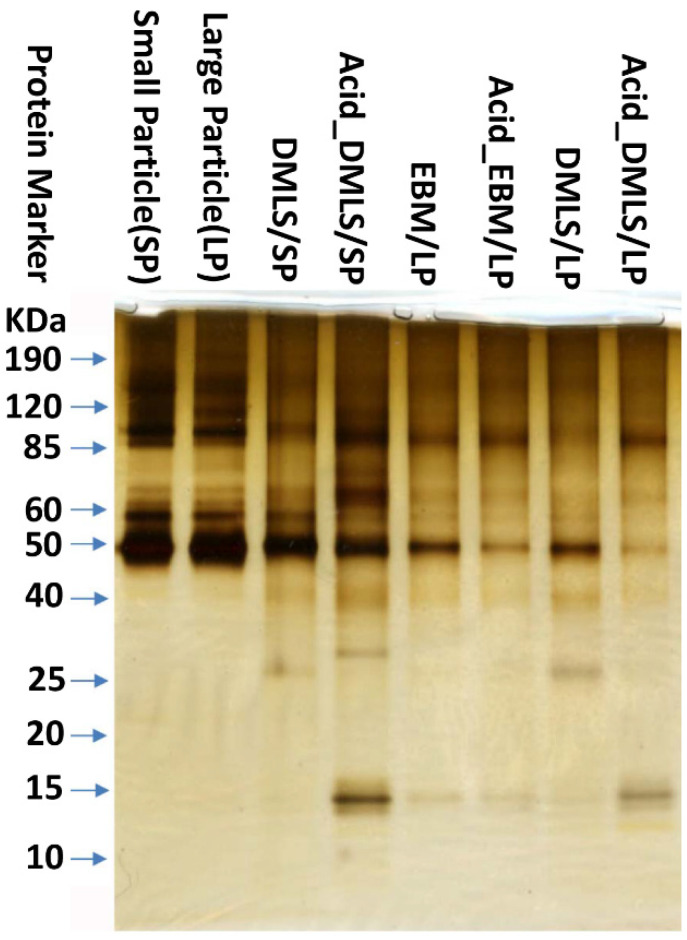
SDS-PAGE analysis of the protein fraction adsorbed on the surfaces of the particles and 3D printed Ti discs. The same stained protein bands above 50 KDa were seen in the small and large particle groups. After 3D printing process, proteins with small molecular weights from 10 to 40 KDa were observed only in the 3D printed discs. There was no difference between EBM/LP and Acid_EBM/LP. The DMLS/SP- and DMLS/LP-printed discs had the stained protein located at around 25 KDa before acid etching. However, after acid etching, the proteins located at around 25 KDa disappeared while the proteins at around 15 KDa appeared. Left panel is protein molecular weight markers in KDa. DMLS: direct metal laser sintering. EBM: electron beam melting. LP: large particle, SP: small particle.

**Figure 5 bioengineering-09-00514-f005:**
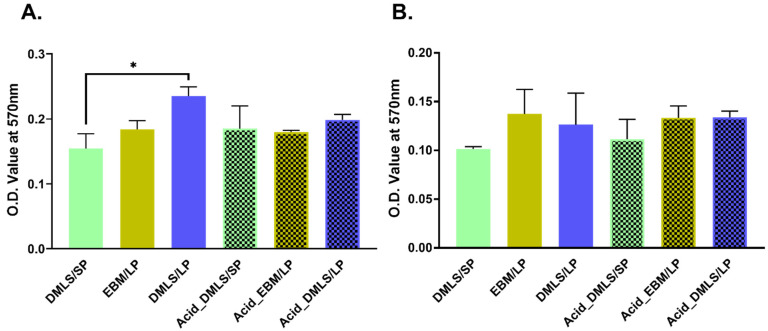
Cell adhesion and proliferation on the surfaces of six types of 3D printed Ti discs were measured by MTT method. (**A**) Cell adhesion at 24 h to the surfaces of six types of 3D printed Ti discs. There was a statistically significant difference found between the DMLS/SP and DMLS/LP groups, which means that more cells attached to the surface 3D printed by laser with the large particles than with the small particles. In addition, there was no difference in cell adhesion before and after acid etching. (**B**) Cell proliferation at 14 days. There were no statistically significant differences among all groups. DMLS: direct metal laser sintering. EBM: electron beam melting. LP: large particle, SP: small particle. * indicates *p* < 0.01; performed in triplicate.

**Figure 6 bioengineering-09-00514-f006:**
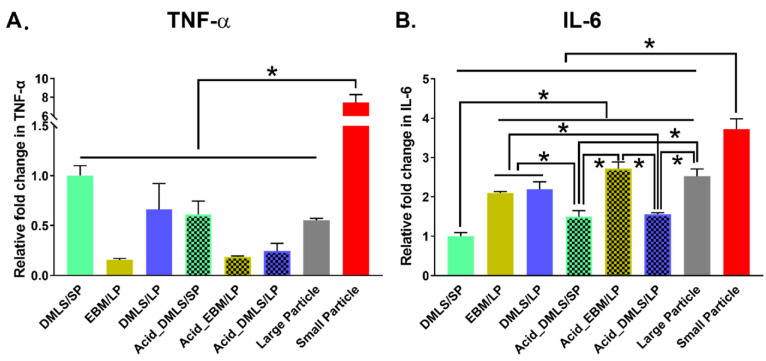
TNF-α and IL-6 genes expression of MC3T3-E1 cells cultured on 6 types of 3D printed Ti discs with small and large particles for 24 h was quantified by real-time qPCR using 2^−^^ΔΔCt^ method. (**A**) TNF-α gene expression. The small particles showed an increased (more than 7 times) in TNF-α gene expression compared to the 3D printed discs as well as the large particles. There was no statistically significant difference among the remaining groups. (**B**) IL-6 gene expression. The small particles showed the highest IL-6 gene expression compared to the 3D printed discs as well as the large particles. The DMLS/SP group had the lowest IL-6 gene expression. Before acid etching, IL-6 gene expressions were higher in the discs printed using EBM/LP and DMLS/LP than DMLS/SP, while there was no difference between EBM/LP and DMLS/LP. After acid etching, IL-6 gene expression was increased in the Acid_DMLS/SP group and decreased in the Acid_DMLS/LP group. The Acid_EBM/LP group had about a 2-fold increase in IL-6 gene expression compared to the Acid_DMLS/SP and Acid_DMLS/LP. There was non-significant difference in EBM discs before and after acid etching. DMLS: direct metal laser sintering. EBM: electron beam melting. LP: large particle, SP: small particle. * indicates *p* < 0.01; performed in triplicate.

**Figure 7 bioengineering-09-00514-f007:**
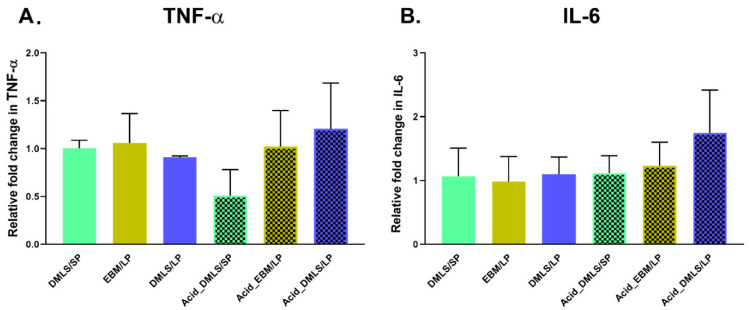
TNF-α and IL-6 gene expressions of MC3T3-E1 cells cultured on 6 types of 3D printed Ti discs at 21 days. (**A**) TNF-α gene expression. There were no statistically significant differences among the 3D printed discs. (**B**) IL-6 gene expression. There were no statistically significant differences among the 3D printed discs. DMLS: direct metal laser sintering. EBM: electron beam melting. LP: large particle, SP: small particle. Performed in triplicate.

**Figure 8 bioengineering-09-00514-f008:**
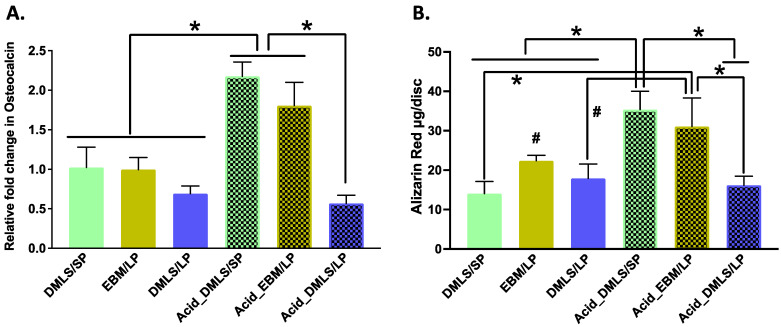
Mineralization of MC3T3-E1 cells cultured on 3D printed Ti discs at 21 days. (**A**) Osteocalcin gene expression by real-time qPCR using the 2^−^^ΔΔCt^ method and normalized against DMLS/SP. The Acid_DMLS/SP and Acid_EBM/LP had the highest osteocalcin gene expressions compared to the other printed discs. There was no statistically significant difference among the rest of the groups. (**B**) The alizarin red content of each printed disc at 21 days, which represents the calcium amount in the mineralization formed by MC3T3-E1 cells on each type of the surfaces of 3D printed Ti discs. The Acid_DMLS/SP and Acid_EBM/LP discs had the highest mineralization and bone nodule formation compared to the other discs. There were no statistically significant differences in alizarin red content among the remaining groups. DMLS: direct metal laser sintering. EBM: electron beam melting. LP: large particle, SP: small particle. * *p* < 0.01, # *p* < 0.05; performed in triplicate.

**Table 1 bioengineering-09-00514-t001:** The assay ID for the probes using for qRT-PCR.

Gene	Assay ID
GAPDH	Mm99999915_g1
TNF-α	Mm00443258_m1
IL-6	Mm00446190_m1
OCN	Mm00649782_gH

**Table 2 bioengineering-09-00514-t002:** EDX analysis for small particles, large particles, and each fabricated Ti disc.

Elements	Atomic Percentage (wt%)							
	SP	LP	EBM/LP	Acid_ EBM/LP	DMLS/LP	Acid_ DMLS/LP	DMLS/SP	Acid_ DMLS/SP
Ti	74.64	78.35	78.8	80.5	81.4	80.4	80.2	78.8
Al	4.12	6.12	5.9	4.2	4.7	4.7	6	6.5
V	3.98	3.91	3.4	3.7	3.6	4	4	3.8
O	6.24	4.18	5.5	4.8	4.1	5.2	4.7	5.3
C	10.53	7.45	6.5	6.8	6.1	5.7	4.3	5.1
Fe	0.5						0.8	0.5

## Data Availability

The original data were available from the corresponding author upon an appropriate request.

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
