# Peer review of "Effects of Different Titanium Surfaces Created by 3D Printing Methods, Particle Sizes, and Acid Etching on Protein Adsorption and Cell Adhesion, Proliferation, and Differentiation"

_bioengineering, 2022, doi:10.3390/bioengineering9100514_

Round 1

Reviewer 1 Report

This study aimed at investigating the effects of different titanium discs prepared from DMLS and EBM on osseointegration, especially discussed the particle size and acid etching on osteoblast differentiation. This research showed that though particles could improve osseointegration effects, but also bring about inflammation and reduce bone formation.

However, to address the osteoinduction properties of the treated titanium surface, the research lacks important control groups, including tissue culture plate and non-treated Ti surface. Moreover, some of the experiment settings are too simple to draw the conclusion, here are the related detailed comments:

1.     Though authors showed the protein bands adsorbed on different discs, it’s also important to show the amount of proteins. In the method, it was showed that 2ug proteins of each group were loaded, however, the bands showed non-equal level of protein intensity.  

2.     Protein adsorption and cell adhesion highly depend on hydrophobicity and roughness, authors should show those properties and discuss. 

3.     Were your cell proliferation assay on different days using the same protocol? If so, why the OD value of day 14 lower than that on day 1? It should be much more cells than on day 1. Otherwise it might indicate that the titanium discs inhibit cell proliferation or viability. 

4.     24 hour is too long for cell adhesion. Cells only take couple of hours to attach to surface. If you want to interpret that the surface affects cell adhesion, you’d better show shorter times. Also, MTT is a cell viability assay, it could reflect cell number in some cases but it cannot represent cell adhesion ability.

5.     It’s better to use a segment Y-axis graph of Figure 4a to compare the TNF-a level for other groups. It looks like the trend is different with IL-6. How should you explain that? TNF-a and IL-6 gene expression, 1 and 21 days

6.     Is that the same data in Fig 5 and day 1 in Fig 6? I don’t think is suitable to put replicate data twice. Also, the statistic analyze is not right. If you want to compare the D1 and D21, you should regard only 1 group as the standard (represents 1) but not the DMLS/SP of D1 and D21 at the same time. It’s useless if you use different standard but put them together as comparison. I recommend remove Figure 6 instead, since it’s incomparable.

7.     When implant was inserted, osteoblast is not the first type of cells that attached to the surface, as well as not the major cell type attributing the inflammation response and release inflammatory cytokines. It’s better to check immune cells response on those surfaces to illustrate the immunogenic properties of different AMs.

Author Response

Reviewer 1

Comments and Suggestions for Authors

This study aimed at investigating the effects of different titanium discs prepared from DMLS and EBM on osseointegration, especially discussed the particle size and acid etching on osteoblast differentiation. This research showed that though particles could improve osseointegration effects, but also bring about inflammation and reduce bone formation.

However, to address the osteoinduction properties of the treated titanium surface, the research lacks important control groups, including tissue culture plate and non-treated Ti surface. Moreover, some of the experiment settings are too simple to draw the conclusion, here are the related detailed comments:

  1. Though authors showed the protein bands adsorbed on different discs, it’s also important to show the amount of proteins. In the method, it was showed that 2ug proteins of each group were loaded, however, the bands showed non-equal level of protein intensity.  

In this study, we focused on protein qualification, but not quantitation. There may be some errors at protein measurement when using BCA Protein Assay or other methods such as Bradford or spectrophotometry. However, such errors do not affect protein qualification result. In other words, through the SDS-PAGE gel, we could identify what proteins were adsorbed on the different surfaces based on protein molecular weight. To examine what proteins they exactly are and how much each protein is attached to different surfaces, it is our next project. We will use protein mass spectrometry to determine what protein each band is and choose the exact protein to study its absorption amount and locate its exact site on the different surfaces of 3D printed discs.

  1. Protein adsorption and cell adhesion highly depend on hydrophobicity and roughness, authors should show those properties and discuss. 

We have added the surface roughness data shown in Figure 3 and EDX analysis data shown in Figure 2 and Table 2. Please see the highlighted parts in Results Section.

  1. Were your cell proliferation assay on different days using the same protocol? If so, why the OD value of day 14 lower than that on day 1? It should be much more cells than on day 1. Otherwise it might indicate that the titanium discs inhibit cell proliferation or viability. 

We apology for the misleading. The proliferation and cell adhesion were separate experiments, cell numbers used in the beginning for these two experiments were different.  The initial cell number in cell proliferation experiment was half (1×105 MC3T3-E1) of the one in cell adhesion (2×105 MC3T3-E1) in order to avoid cell confluence in titanium surfaces during the 14 days culture period.

  1. 24 hour is too long for cell adhesion. Cells only take couple of hours to attach to surface. If you want to interpret that the surface affects cell adhesion, you’d better show shorter times. Also, MTT is a cell viability assay, it could reflect cell number in some cases but it cannot represent cell adhesion ability.

We totally agree with the reviewer in this point, specifically when treated cell culture plates are used and 4 to 5 hours are usually sufficient for cell attachment. In our case, we used titanium. Compared to culture dish, adhesion of the cells to titanium surface during 4-5 hours was weak. After PBS rinse, almost all the cells were removed from the surface. We found 24 hours was a good time point to compare cell adhesion to the different fabricated Ti scaffolds. Our observations are in concordance with previous works investigating cell adhesion on titanium surfaces. Reference added to the manuscript: Resende, Cristiane X. (2010). "Cell adhesion on different titanium-coated surfaces". MATERIA-RIO DE JANEIRO (1517-7076), 15 (2), p. 420.

Because of the observed delay of MC3T3-E1 cell to attach to the fabricated Ti discs, we assumed that MTT test will be adequate to quantify the cell number on different surfaces as previously described: Miki I, Ishihara N, Otoshi M, Kase H. Simple colorimetric cell-cell adhesion assay using MTT-stained leukemia cells. J Immunol Methods. 1993 Sep 15;164(2):255-61.

  1. It’s better to use a segment Y-axis graph of Figure 4a to compare the TNF-a level for other groups. It looks like the trend is different with IL-6. How should you explain that? TNF-a and IL-6 gene expression, 1 and 21 days

We changed Y-axis from linear to segment in Figure 6a.

We agree with the reviewer that TNF-α and IL-6 gene expressions have different trends. This may be explained by the high Ct values for both IL-6 and TNF-α when compared to the house keeping gene, which indicate that these 2 genes were not highly stimulated when MC3T3-E1 cells were cultured on Ti discs. An external stimulation of both IL-6 and TNF-α by LPS will show more accurate response of MC3T3 cultured on Ti discs to inflammation. The only obvious observation is that the IL-6 expression at 24h shows the same trend as the surface roughness measurements shown in Figure 3. Higher roughness in EBM/LP and Acid_EBM/LP correlate with the highest IL-6 expression measured by qPCR. Adsorbed proteins on the different surfaces may also contribute to this difference in the expression of both IL-6 and TNF-α at day 1 and day 21, please see yellow highlights in Discussion Section.

  1. Is that the same data in Fig 5 and day 1 in Fig 6? I don’t think is suitable to put replicate data twice. Also, the statistic analyze is not right. If you want to compare the D1 and D21, you should regard only 1 group as the standard (represents 1) but not the DMLS/SP of D1 and D21 at the same time. It’s useless if you use different standard but put them together as comparison. I recommend remove Figure 6 instead, since it’s incomparable.

We agree with the reviewer. We removed the old Figure 6.

  1. When implant was inserted, osteoblast is not the first type of cells that attached to the surface, as well as not the major cell type attributing the inflammation response and release inflammatory cytokines. It’s better to check immune cells response on those surfaces to illustrate the immunogenic properties of different AMs.

We completely agree with the reviewer that other inflammatory cells such as neutrophils and macrophages are the first cells in contact with Ti implants. Also, the interaction between the above-mentioned immune cells and different surface types resulting from different AMs method is not well investigated. Since Ti implants are biocompatible and well tolerated with the body’s immune system, our main focus in this study, was to determine which AM Ti surface is optimal to bone formation or osseointegration. Therefore, we focused on the effects of 3D printed Ti different surface properties on the pre-osteoblast cell adhesion, proliferation and differentiation.

Reviewer 2 Report

The authors  studied the different titanium surfaces, created by AM methods 

The manuscript is very interesting for those who work with biomaterials in orthopedy and dentistry and of course for materials scientists, who manufacture AD  SLM or EBM parts of implants.

However, some data are missing  

Firstly how was the initial powder of Ti6Al4V protected against oxidation, since it is generally known that titanium forms in the air or in liquids in a few milliseconds very dense  6-8 nm thick thin film of TiO which protect the material against further corrosion. 

Is there any difference between the two procedures DSLM and EBM?

Please describe the influence of acid treatment on this protective film. 

Also, data on surface properties of 6 different samples, roughness, wettability, and surface chemistry by XPS n or ToF SIMS are missing.

Author Response

Comments and Suggestions for Authors

The authors  studied the different titanium surfaces, created by AM methods. The manuscript is very interesting for those who work with biomaterials in orthopedy and dentistry and of course for materials scientists, who manufacture AD  SLM or EBM parts of implants.

However, some data are missing  

Firstly how was the initial powder of Ti6Al4V protected against oxidation, since it is generally known that titanium forms in the air or in liquids in a few milliseconds very dense  6-8 nm thick thin film of TiO which protect the material against further corrosion. 

Is there any difference between the two procedures DSLM and EBM?

Please describe the influence of acid treatment on this protective film. 

We purchased commerical Ti6Al4V (grade 5) powder from 3D Systems. Ti6Al4V (grade 5) have a chemical composition that meets the requirements of ASTM B265, B348 (grade 5), F2924, F3302, ISO 5832-3 and Werkstoff Nr. 3.7165.These powders were stored in the air prior to use for 3D printing.

The differences between the two procedures DMLS and EBM are energy source, working enviroments and powder preheating requirement. Please see the higelighted yellow part in Materials and Methods Section.

Also, data on surface properties of 6 different samples, roughness, wettability, and surface chemistry by XPS n or ToF SIMS are missing.

We have added the surface roughness data shown in Figure 3 and EDX analysis data shown in Figure 2 and Table 2. Please see the highlighted parts in Results Section. Unfortunately, we were not able to run XPS or ToF SIMS analysis on the samples.

Round 2

Reviewer 1 Report

Thank you for the response. The manuscript improves a lot after the revision. However, there are still some minor flaws need to be corrected:

1. Please double check your spelling on special characters. For example line 158, "1x105 cells", line 202 and line 187 (2-∆∆Ct) should be superscripts. Line 206 "1x"should not be letter x. Figure 8B y axis title should be "μg", also that graph has a weird box line.

2. I did not see changes on figure 6a.

Author Response

Thank you so much for reviewing the manuscript and for your valuable comments.

  1. Please double check your spelling on special characters. For example line 158, "1x105 cells", line 202 and line 187 (2-∆∆Ct) should be superscripts. Line 206 "1x"should not be letter x. Figure 8B y axis title should be "μg", also that graph has a weird box line.

We fixed all typo/spelling, formating for special characters.

2. I did not see changes on figure 6a.

Sorry for missing that, we uploaded the figure 6a with segmented Y-axis.

Reviewer 2 Report

The authors  improved the text of the manuscript sufficiently 

but there are missing data of Ti6Al4V powder exposure to the air before  EBM and DSLM  - 

 from my point of view, it is hard to compare the roughness of EBM and DSLM because of the different granulation  DSLM (5-40 microns) and (45-150 microns) and EBM 100 microns. please explain in the text 

Please add the data of the Ti6Al4V exposure to the air before  the DSLM and EBM

Author Response

Thank you so much for reviewing the manuscript and for your valuable comments. 

The authors  improved the text of the manuscript sufficiently but there are missing data of Ti6Al4V powder exposure to the air before  EBM and DSLM  -  from my point of view, it is hard to compare the roughness of EBM and DSLM because of the different granulation  DSLM (5-40 microns) and (45-150 microns) and EBM 100 microns. please explain in the text. Please add the data of the Ti6Al4V exposure to the air before the DSLM and EBM.

We added the EDX data of Ti6A14V powder exposure to the air before EBM and DSLM. Please see the updated Figure 2 and Table 2. 

We added one­­­­ sentence about surface roughness under Results Section (highlighted in yellow):

"This roughness measurement should be interpreted carefully because the roughness Ra came from both the surface roughness and the fused particle height."